# Factors affecting motivation of close-to-community sexual and reproductive health workers in low-income urban settlements in Bangladesh: A qualitative study

Ilias Mahmud[1,2]*, Sumona Siddiqua[2], Irin Akhter[2], Malabika Sarker[2,3], Sally Theobald[4], Sabina Faiz Rashid[2]

1 Department of Public Health, College of Public Health and Health Informatics, Qassim University, Al Bukairiyah, Saudi Arabia, 2 BRAC James P Grant School of Public Health, BRAC University, Mohakhali, Dhaka, Bangladesh, 3 Heidelberg Institute of Global Health, Heidelberg University, Heidelberg, Germany, 4 Department of International Public Health, Liverpool School of Tropical Medicine, Pembroke Place, Liverpool, United Kingdom

* i.emdadulhaque@qu.edu.sa

**Data Availability Statement:** Data for this study are qualitative and contain sensitive information. Therefore, the data cannot be openly published.

## Abstract

Close-to-community (CTC) health workers play a vital role in providing sexual and reproductive health services in low-income urban settlements in Bangladesh. Retention of CTC health workers is a challenge, and work motivation plays a vital role in this regard. Here, we explored the factors which affect their work motivation. We conducted 22 in-depth interviews in two phases with purposively selected CTC health workers operating in low-income urban settlements in Dhaka, Bangladesh. We analyzed our data using the framework technique which involved identifying, abstracting, charting, and matching themes across the interviews following the two-factor theory on work motivation suggested by Herzberg and colleagues. Our results suggest that factors affecting CTC sexual and reproductive health workers' work motivation include both extrinsic and intrinsic factors. Extrinsic or hygiene factors include financial incentives, job security, community attitude, relationship with the stakeholders, supportive and regular supervision, monitoring, and physical safety and security. While, the intrinsic factors or motivators are the perceived quality of the services provided, witnessing the positive impact of the work in the community, the opportunity to serve vulnerable clients, professional development opportunities, recognition, and clients' compliance. In the context of a high unemployment rate, people might take a CTC health worker's job temporarily to earn a living or to use it as a pathway move to more secure employment. To maintain and improve the work motivation of the CTC sexual and reproductive health workers serving in low-income urban settlements, organizations should provide adequate financial incentives, job security, and professional development opportunities in addition to supportive and regular supervision.

This is a requirement from the ethics committee we received approval from. Please contact the Co-Chair of the IRB (irb-jpgsph@bracu.ac.bd) for permission to access data. Data are also available from the PI (sabina@bracu.ac.bd) on request. However, approval must be obtained from the Institutional Review Board (IRB) of the BRAC James P. Grant School of Public Health.

**Funding:** This research was nested within the REACHOUT consortium, funded by the European Union FP7 grant (number 306090). The funders had no role in study design, data collection and analysis, decision to publish, or preparation of the manuscript.

**Competing interests:** The authors have declared that no competing interests exist.

**Abbreviations:** CTC, Close-to-Community; SRH, Sexual and Reproductive Health; CHW, Community Health Worker; LMICs, Low and middle-income countries; NGO, Non-Governmental Organization.

# Background

More than 35% of the urban population of Bangladesh lives in low-income, and often informal, settlements [1]. The population in these settlements is growing by 7% annually, mainly because of an influx of rural migrants [1]. Current public health services are inadequate to satisfy the healthcare needs of this growing population. This vacuum in health services is mainly filled by informal and poorly regulated private health service providers with dubious quality [2]. However, a few non-government organizations (NGOs) also operate in these settlements through Close-to-Community (CTC) health workers [2]. CTC health workers are a group of heterogeneous providers of health services at the community level and known by different titles in different country contexts, such as community health workers, community health volunteers, community distributors, community health nurses, community-directed health workers, health auxiliaries, health promoters, family welfare educators, village health workers/volunteers/teams, community health aides, barefoot doctors, traditional healers, practitioners who combine traditional and modern medicine, allopathic practitioners, drug sellers and faith healers [3].

CTC health workers play a vital role in achieving universal health coverage. They are instrumental in providing cost-effective and efficient delivery of primary healthcare in low- and middle-income countries (LMICs) [4–6]. They link the formal health systems to the community that otherwise would remain beyond the reach of formal health facilities [7]. CTC health workers, both formal and informal, are an important part of our health system in mitigating the challenges of the shortage of health professionals [8]. Yet, they remain particularly vulnerable to de-motivation and, consequently, high rates of attrition [9,10]. This puts a considerable strain on our health systems, due to the cost and time necessary to recruit and train new health workers [11].

Motivation is an important factor in preventing health worker attrition [12,13], burnout [14], turnover [15], and promoting better quality services [16]. The motivation of the CTC health workers is associated with both individual and programme performance [17,18]. Understanding the factors affecting the motivation of CTC health workers is important because community health programmes' success and failure depend on their performance [19]. Effective management, training opportunities, and supportive supervision along with both financial and non-financial incentives are needed to keep them motivated, performing, and retained for a longer term [20,21].

CTC Sexual and Reproductive Health (SRH) workers provide promotional, preventive, and curative SRH services, including menstrual regulation, in low-income urban settlements in Bangladesh and experience even greater challenges because of the sensitivity of SRH issues [22]. Hence, it is particularly important to understand the factors which motivate or demotivate them to continue their job for the improvement of community SRH services in low-income urban settlements. This paper reports the factors affecting the motivation of CTC SRH workers operating in low-income urban settlements in Bangladesh.

# Methods

This study is a part of a larger project on improving CTC SRH services in low-income urban settlements in Bangladesh [22] and relied on a qualitative approach. We conducted 22 in-depth interviews with purposively selected CTC SRH workers (Table 1) using semi-structured interview guidelines. CTC SRH workers we interviewed included 14 salaried workers and eight volunteers who receive a little monetary incentive based on their performances. We recruited CTC SRH workers from both the NGO and Government sectors. We conducted our in-depth interviews in two phases. In the first phase, we interviewed 10 CTC SRH workers and

**Table 1. Characteristics of the participants.**

| No | Age | Gender | Designation | Job-status | Job description | Experience | Educational attainment |
|---|---|---|---|---|---|---|---|
| 1. | 27 | Female | Health Educator | Salaried | Door-to-door visits, pharmacy, diagnostic center, and other clinic visits, ANC, PNC, and MR counseling and referral, inform community people about satellite clinics and note their regular work in the official register. Supervised by the Clinic Manager | 4 months | Masters |
| 2. | 45 | Male | Field Coordinator | Salaried | Pharmacy, diagnostic center, and other clinic visits, ANC, PNC, and MR counseling and referral, maintain their regular work in the official register. Supervised by the Clinic Manager. | 3 years | Masters |
| 3. | 32 | Male | Health Educator | Salaried | Previously mentioned | 1 year 6 months | Masters |
| 4. | 28 | Male | Outreach worker | Salaried | Door-to-door visits, pharmacy, diagnostic center, and other clinic visits, ANC, PNC, and MR counseling and referral, inform community people about satellite clinics, note their regular work in the official register, and supervise volunteers. Supervised by a service promoter. | 6 months | Masters |
| 5. | 32 | Female | Marie Stopes Volunteer | Volunteer | Attending satellite clinics and counseling community people for taking services from satellite clinics. Refer people to satellite clinics and static clinics. Supervised by an Outreach Worker. | 4 months | Primary |
| 6. | 32 | Male | Field Coordinator | Salaried | Previously mentioned | 8 years | Bachelor |
| 7. | 50 | Female | Marie Stopes Volunteer | Volunteer | Previously mentioned | 6 months | No formal education |
| 8. | 33 | Female | Service Promoter | Salaried | Door-to-door visits, pharmacy, diagnostic center and other clinic visits, ANC, PNC, and MR counseling and referral, inform community people about satellite, note their regular work in the official register, supervise Outreach Worker and MSV and supervised by a Clinic Manager. | 5 years | Masters |
| 9. | 22 | Female | Service Promoter | Salaried | Previously mentioned | 2 years 6 months | Higher Secondary |
| 10. | 44 | Female | Service Promoter | Salaried | Previously mentioned | 19 years | Bachelor |
| 11. | 22 | Female | Peer Educator | Volunteer | Attend community awareness-raising programmes, peer education, and referral. | 4 months | Primary |
| 12. | 35 | Female | Peer Educator | Volunteer | Mentioned previously | 1 year | No formal education |
| 13. | 31 | Female | Paramedic | Salaried | Give services and counseling for general health problems, ANC, PNC, MR, PAC & normal delivery at static clinic and satellite clinic. Supervised by a Clinic Manager | - | Secondary |
| 14. | 30 | Female | Community Health volunteer | Volunteer | During monthly household visits, they provide health promotion sessions and educate families on nutrition, safe delivery, family planning, immunizations, hygiene, and water and sanitation. They also use this time to sell health products, such as basic medicine, sanitary napkins, and soap. | - | Primary Incomplete |
| 15. | 43 | Female | Family welfare visitor | Salaried | Give general health and family planning services at union health and family welfare centers throughout Bangladesh | 2 years | Higher Secondary |
| 16. | 40 | Female | Family welfare visitor | Salaried | Mentioned previously | - | Masters |
| 17. | 42 | Female | Family Welfare Assistant | Salaried | Trained to screen and offer a variety of services including family planning, support for pregnant women, TB case detection and treatment support, and immunizations among others. | - | Secondary |
| 18. | 45 | Female | Community Health volunteer | Volunteer | Previously mentioned | 2 years | Primary |
| 19. | 38 | Female | Community Health volunteer | Volunteer | Previously mentioned | - | No formal education |
| 20 | 25 | Female | Community Health volunteer | Volunteer | Previously mentioned | - | Higher Secondary |
| 21. | 25 | Female | Family welfare visitor | Salaried | Mentioned previously | - | Secondary |
| 22. | 41 | Female | Family welfare assistant | Salaried | Mentioned previously | - | Secondary |

in the second phase, we did another 12 interviews. First-phase interviews were intended to understand the overall challenges of the CTC SRH workers including their motivation. While the second phase of interviews exclusively explored the factors affecting their motivation. We have conducted interviews at the participants' residences or workplaces based on their convenience.

All interviewers had a university degree in a social science discipline with prior experience in qualitative research. In addition, they received training on qualitative interviews and ethical issues specific to this research. Interviews were conducted in the local language (Bangla) and were recorded with an audio recorder. All the participants gave their written informed consent for the interviews.

Interviews were transcribed verbatim in Bangla by the respective interviewers. All transcripts were then translated into English by a bilingual professional translator with prior experience in translating qualitative interviews. The authors read each interview transcript several times to identify the factors positively or negatively affecting the motivation of CTC SRH workers. We used the framework technique to analyze our data [23]. This technique involved identifying, abstracting, charting, and matching themes (in our case, factors affecting the motivation of CTC SRH workers) across the interviews following the two-factor theory on work motivation suggested by Herzberg and colleagues [24]. All identified statements were considered, but statements were consolidated either into intrinsic factors, also known as motivators, or extrinsic factors, also known as hygiene factors, as Herzberg and colleagues suggested in their two-factor theory on work motivation [24] where it was perceived that the underlying factors were the same. The intrinsic factors or motivators are associated with the need for growth or self-actualization. These factors included achievement, recognition, the work itself, responsibility, advancement, and the possibility for growth. Whereas the extrinsic factors or hygiene factors are related to the need to avoid unpleasantness and included company policies and administration, relationship with supervisors, interpersonal relations, working conditions, and salary [24].

## Results

To explore the factors which affect the motivation of CTC SRH workers in low-income urban settlements in Bangladesh, we employed a qualitative approach to data collection and analysis. Our findings suggest that the motivation of CTC SRH workers is influenced by both extrinsic and intrinsic factors (Table 2).

**Table 2. Empirical constructs of motivation in CTC SRH workers in low-income urban settlements in Bangladesh.**

| Extrinsic/hygiene factors | Intrinsic factors/motivators |
|---|---|
| **Community factors** | **Perceived value in the work** |
| Community attitudes toward CTC SRH workers | Perceived quality of the SRH services provided |
| Relationship between CTC SRH workers and the community | Opportunity to serve vulnerable women |
| Perceived safety and security in the community | Perceived impact of the work on clients' lives |
| **Organizational factors** | **Professional growth** |
| Relationship with co-workers and supervisors | Opportunity to improve knowledge and skills |
| Supervision | Promotion opportunities |
| Workload/target | |
| **Financial stability factors** | **Recognition** |
| Remuneration | Recognition from the community, clients |
| Job security | Recognition from supervisors |
| Retirement benefits | Clients' compliance |

## Extrinsic factors

Extrinsic or hygiene factors, we identified, are grouped under community, organizational and financial stability factors.

### Community factors

Community factors included community attitudes towards CTC SRH workers, the relationship between CTC SRH workers and the community, and perceived safety and security in the community.

**Community attitudes and relationship with the community.**   CTC SRH workers mostly reported that the community has positive attitudes towards them and their services. They appreciate the need of having positive community attitudes in carrying out their job duties. Hence, they put a conscious effort into developing and maintaining positive community attitudes toward them and their activities. In this regard, a CTC SRH worker told, *"most of the people, almost 90% of them, behave well."* (Field coordinator, male, 35-year-old). While another further elaborated and said, *"Local people have positive attitudes toward me. For example, when I walk through the road, they catch me and start to talk about their problems, or just start social chatting. Sometimes, I am time pressured, but I must talk to them. I feel, if I do not hear them today, they will not come to me tomorrow for my services; they will go elsewhere. That's why I pay attention to them [even when I am busy]."* (Family welfare visitor, female, 43-year-old).

The job of CTC SRH workers requires them to meet and interact with people in the community. Some workers find it enjoyable. For example, a health educator said, *"I am satisfied [with my job]. I like talking to people. . . [Because of my job] I can meet new people and get acquainted with them. Sometimes, I find people who have migrated from my village"*. (Female, 27-year-old).

On contrary, CTC SRH workers also experience negative behaviours from the community people which demotivates them. Often this is because of the stigma associated with SRH issues. For example, a female volunteer said *"Many do not like us discussing SRH issues with their adolescent children. Sometimes some people misbehave with me when I visit their house. I feel sad at that time.'* (Community health volunteer, 32-year-old).

**Perceived safety and security in the community.**   Sometimes, CTC SRH workers are concerned with their safety in the field. They feel the need for a partner while working in the community. Concern with own safety works as a demotivating factor. In this regard, a female service promoter said, *'working alone is a boring thing. If I get a man or woman work partner, it will help me to do my work better and increase my client number. Moreover, working alone is risky for women. In some areas, people are not good. Sometimes/some places, where a man can enter, a woman cannot. In this case, if someone accompanies me, I feel secure, and it also helps to do my work better"*. (Service Promoter, 22-year-old). Safety is not a concern only for female workers. A male health educator said, *"The most challenging is slum [informal settlements] visits. I can enter a pharmacy at any time, but I cannot enter a slum at any time. . .I am a man; it is risky for me to enter a slum at any time. It is risky for women as well.'* (Health educator, 32-year-old).

### Organizational factors

**Relationship with co-workers and supervisors.**   Most of the CTC SRH workers, we interviewed, mentioned the working environment as good and motivating. They were coming timely to the office and going to the field regularly. They like to have a good bonding with co-workers, and openly discuss their field experiences, problems, comforts, and discomforts with them. One participant stated, *"There are seniors and juniors, but we all are like family here. I*

*can talk to everyone openly, and pleasantly. This is a good environment. Our supervisors behave well with us and honour us. We also respect them. We have noticed that in some organizations it is necessary to address your supervisor as boss or sir. Here, we address them as brothers, and they also address us as brothers. So, a cordial work environment is developed".* (Field coordinator, male, 35-year-old).

Another CTC SRH worker said, *"All of my colleagues are good. We help each other in work. We can discuss with each other when there are issues [challenges], even personal family issues. Therefore, we get motivation in our work."* (Health educator, female, 27-year-old).

We found that CTC SRH workers help each other when they experience challenges. They help each other to achieve their monthly job target. This support and cooperation between them work as a motivating factor. A CTC SRH worker mentioned, *"I like the environment. I like the cooperation I receive from my colleagues. I like their behaviour. That's why I like to work here."* (Health educator, male, 32-year-old). In this regard, another told that "*Reza [pseudonym] is a very good colleague. When we are short of the daily target [number of clients], he tries to work harder and say, "Let's go sister; let's visit more. Today the number of clients is less."* (Volunteer CTC SRH worker, female, 50-year-old). Therefore, CTC workers work better due to the friendly and supportive working environment. Appreciation is one of the key factors to enthuse to work better.

**Supervision.** We found that supportive supervision and regular supervision motivate CTC SRH workers. They reported that their supervisors always help them when they face difficulties in the field. They immediately contact their supervisors when they need their suggestions and decisions. The data suggest that supportive and regular supervision helps in three ways- first, by improving CTC SRH workers' skills, second, through supportive supervision supervisors can be positive role models and third, as a tracking mechanism of the performance of the CTC SRH workers.

CTC SRH workers reported that their skills have improved following supportive and regular supervision. Therefore, they feel good and motivated when they receive supportive and regular supervision. A CTC SRH worker mentioned, *"I did not understand my work well before. Because of her supervision, it became possible for me to take my work easily". (Service promoter, female, 22-year-old).* Another CTC SRH worker, who joined her job 7 months ago, mentioned, *"She [my supervisor] is teaching me how to do my work better as I'm new. She is in the work for a long. She teaches me well and tells me ways of getting success in my job. This is how I get motivation from her. My previous supervisor was also very good. She used to visit the field with me and used to correct me when I did any mistake while talking to people. She used to guide us to better attract the attention of people through our talk".* (Health educator, female, 27-year-old).

We also found that CTC SRH workers get inspired and motivated when their supervisor accompanies them in the field. It helps them to take instant decisions like providing subsidies to a patient in an emergency. Sometimes, local people create pressure on the CTC SRH workers to stop providing sensitive SRH services, such as menstrual regulation. In these cases, their supervisors can intervene during field visits- they talk to the local people including gatekeepers and politicians to solve the problems. Working in the field for a stigmatized issue like menstrual regulation is challenging. Hence, CTC SRH workers sometimes feel neglected when their supervisors do not visit the field because of their other workload.

We further found that regular supervision keeps the CTC SRH workers on track and motivates them to perform well, hence is appreciated. In this regard, a CTC SRH worker said, "*. . .if they leave us un-monitored and don't keep any track, there will be a tendency of lagging behind. But, if the work is monitored regularly, for example, if my work is supervised everyday then a sense of accountability will work in me and this sense of accountability will motivate me in my work.* (Health educator, female, 27-year-old).

On the contrary, we found that unsupportive supervision demotivates CTC SRH workers. For example, a health promoter mentioned, *"I feel constant pressure from all around me, but still, I work very hard with a smile on my face. . .a few days ago our line manager insulted me in front of others. I was telling [people from other organizations] that I got this big opportunity as a women leader. He [my line manager] stopped me and said, 'what women leadership are you talking about; you have done this and that wrong'. He could have spared me from this insult. He could have provided me feedback in private as he is my supervisor."*

In another instance, the same worker said: *"Well, friendliness is the main thing, not reproaching. My supervisors do not think of it like that. . .I have seen many times that many managers severely reproach supervisees even if they have done good work. They constantly criticize them. Criticism cannot make one skilled or stronger, but motivation can. . . I think the supervision process can improve skills. . . Does supervision mean controlling, no!"*

CTC workers prefer to be monitored based on meeting the targets. However, often they are monitored on maintaining office hours which works as a demotivating factor for some. For example, a community SRH worker mentioned, *"Well when they scold me, I feel demotivated. When they tell me 'You are avoiding work too much; stop doing it.' I become extremely careless sometimes (laughs). Sometimes, I avoid the afternoon session because of my other involvements. They criticize me a lot for this. I wish I could tell them that it does not matter how long I have been in the field; check if I have done the work, check if there is any achievement or not."*

**Workload/Target.** There are many drug stores and private clinics, which are often illegal, poorly regulated, and provide unsafe services, available in and around the community. CTC SRH workers face fierce competition from informal providers, such as drug store salespeople and unregulated or poorly regulated private clinics. In this regard, a worker said, *"Motivation increases when we get clients. It is normal to feel bad if we roam around in the community the whole day but can't find a single client. If the target of ligation and other contraceptive methods can be met, we feel good. We feel bad if we can't meet the target [set by the management]."* (Family welfare assistant, female, 42-year-old). Even after putting in their best effort, when they receive negative feedback from their superiors for not meeting the target, they feel frustrated and demotivated. For example, a worker mentioned, *"I feel tired, demotivated, and depressed. I feel bad. It is felt worse if the target is not achieved even after doing so much hard work. On the other hand, the office also asks, what are you doing?"* (Outreach worker, male, 28-year-old).

Unrealistic targets, pressure to achieve the target and lack of appreciation from the superior for the effort community workers are putting in may result in poor job satisfaction and demotivation. A service promoter mentioned, *"I work hard. I must feel bad when you do not appreciate my effort. I feel constant pressure from all around me."* (Service promoter, female, 33-year-old).

### Financial stability factors

**Remuneration.** Lack of perceived fair remuneration is one of the major factors of demotivation among the CTC SRH workers in informal urban settlements in Bangladesh. This is illustrated by the following quote: *"I am not satisfied with my salary. Besides, annual increment also depends on salary. Annual increment would be less if the salary is less." (Field coordinator, male, 35-year-old).* Another outreach worker said: *"Encouragement is necessary for doing work. The amount they give us as salary is making it next to impossible to live in Dhaka."* (Outreach worker, male, 28-year-old). Echoing the same a female volunteer said: *"I like this work; hence I am doing this. . . But our financial incentives are not justifiable considering our workload. . .If someone asks us about our financial benefits, we feel ashamed. They ask us "why you are doing this work with such a low salary".*

In this regard, a field coordinator said: *"No benefits [increment or bonus] . . .it does not matter how well I work; how much I achieve. Even when I achieve my targets–there are no extra benefits [financial]."* (Field coordinator, male, 32-year-old).

**Job security and retirement benefits.** Lack of job security and retirement benefits may demotivate CTC SRH workers in the low-income urban settlements in Bangladesh. This is evident in the following quotation: *"Many [office-based workers] are appointed as a permanent employee after two, three or four years of service. I am working here for eight long years*! *They are not appointing us as permanent staff. If I leave my job now, I will have to go with an empty hand, life is zero."* (Field Coordinator, Male, 32-year-old).

## Intrinsic factors or motivators

**Perceived value in the work.** CTC SRH workers work in a pluralistic environment where there is fierce competition between different formal, including NGO and private providers, often with dubious quality, and informal providers of SRH services. This context makes the women in need of any sensitive SRH services vulnerable to unsafe practices and exploitation. Our data suggest that knowing that they are providing quality services to vulnerable women and observing the positive impact of their work on vulnerable clients' lives inspire motivation among the CTC SRH workers.

**Perceived quality of the SRH services provided.** Our data suggest that the CTC SRH workers feel motivated when they perceive that the services their organization is providing are of good quality, hence helping people. For example, a health educator mentioned, *"I like helping people. The SRH services we provide here are of good quality. I like this very much. Therefore, I like working here"*. (Female, 27-year-old).

We also found that often CTC SRH workers themselves do not endorse certain contraceptive methods considering the perceived negative effect of such methods. Hence, they find that task demotivating. For example, the same worker mentioned, *"Sometimes there are IUD camps, and we must bring clients for IUD camps. But some of them [women in the community] don't want IUD. They don't like it. I too don't feel good doing that. I feel that it would be better if we don't have to do this. . .It makes problems, and many of them faced problems. It makes some of them bleed very much. As it remains for 10 years, so it can create problems. . . It has even caused the death of some people. Many people fear this contraceptive method as it can bring either a positive or a negative result. People are worried about the negative aspect. So, they are not going for it."*

**Opportunity to serve vulnerable women.** We found that a feeling of satisfaction in serving vulnerable women keeps CTC SRH workers motivated to perform their job responsibilities. This is illustrated by the following quote: *"Everyone doesn't get the opportunity to help people. I am lucky that God has given me this opportunity [to provide SRH services to vulnerable women]"*. (Service promoter, female, 22-year-old).

Another worker told, *"Well, a helpless girl, who cannot tell anyone that she has got pregnant, feels happy when she sees us. I feel good doing these types of work. I mean she cannot tell anyone but feels good to see us. These are the secrets [of my motivation]."* (Service promoter, female, 44-year-old). Another said, *"If we can stand by a patient of our locality, isn't it a matter of joy for us; our encouragement?"* (Peer educator, female).

**Perceived impact of the work on clients.** Our data also suggest that perceiving a positive impact of their work on the community and themselves keeps the CTC SRH workers motivated to carry out everyday job responsibilities. They like to work in low-income urban settlements to promote safe SRH services. Their main job responsibilities included identifying women in need of SRH services including menstrual regulation, providing SRH information

related to prevention and treatment, linking them to a formal clinic, and following up and encouraging the women to receive care in a formal health facility. CTC SRH workers move around the community and spend a great deal of time in the community. Hence, they understand the vulnerability of women in need of menstrual regulation or other SRH services. Therefore, they feel good knowing that they are doing something good for these helpless women. One CTC SRH worker told, *"I am satisfied with my job. I never feel bad to work. I like to serve people. This is a valuable service- extending help to others in great need. People pray for me. I can approach them easily. Many people love me, like me, trust me. When they share their heart feelings with me considering me a near one of their lives, I feel honoured.* (Family welfare visitor, female, 43-year-old).

Helping women for receiving safe and affordable SRH services, particularly menstrual regulation, give these CTC health workers a sense of satisfaction. A female CTC SRH worker said, *"I try my best to help them [women in need of menstrual regulation]. I wish they come to me first instead of wasting thousands of taka [Bangladesh currency; in informal or poor-quality private clinics] by the influence of the middlemen."* (Service promoter, 22-year-old).

### Professional growth

**Opportunity to improve knowledge and skills.**   CTC SRH workers value the opportunity to learn new things after coming into this profession. They learn from the doctors, other health workers of the NGOs, or any partner organizations they are working with. In this regard, a field coordinator said *"I have learned everything after coming here; that's why I am satisfied. I am learning many new things while serving clients [about their SRH issues and available treatment options]"*. (Male, 45-year-old).

Another worker said, *"I don't become upset when they [supervisors] tell me to do things differently because they are teaching me. . .. They are instructing me because I don't know enough. I have the desire to learn. I must learn this. I perform my duties at my best. I would be grateful [to my supervisors] if they could teach me completely and if I could learn from them completely."* (Volunteer, female, 50-year-old).

**Promotion opportunities.**   Promotion opportunities motivate CTC SRH workers. For example, a newly appointed, working only for four months, CTC SRH worker was expecting a promotion following a show of good work: *"It is great if I can do my work well. I would get a promotion if I can fulfill my job responsibilities"*. However, few workers reported that they don't get any promotions and are tired of working in the same position. For example, a health educator mentioned, *"The major point is that no one wants to work at a lower position for a long time. . . For this [lack of promotion opportunity] it may not be possible for me to continue. If I get a better chance and get a good job, I will leave. Everyone wants quality, good position, and respect."* (Health Educator, Female, 27-year-old).

**Professional networking.**   Many community SRH workers have little formal education. However, their job enables them to make connections with doctors working in different hospitals and clinics in and around their neighbourhood. They often use this connection to take their clients to clinics or hospitals and negotiate a reduced service charge for them. In return, they earn respect from the community they are working for. This is hugely motivating for the community SRH workers. For example, a peer educator mentioned, *"I have taken quite a few patients there [a government hospital] . . . There was a woman nearby whose parents have died. She was very poor. Her delivery was due. She had two children, but both died. I knew Mitford [a government hospital], so I took her there. I told the doctors there that she is very poor; her two children have died before; please take less money from her if you can. They gave her all the medicines she needed, and her baby was delivered there. It did not cost her much".* (Peer educator, female, 35-year-old).

## Recognition

**Recognition from supervisors.** When supervisors praise their work, CTC SRH workers feel very inspired. For example, a CTC worker told, *"Supervisor compliments us if he likes our work. Then, he advises us on how to work better by improving our skills."* (Paramedic, female, 25-year-old). In this regard, another motivated CTC worker said, *"We have the freedom. We work the way we want to. I do every work in my responsibility"*. (Family welfare assistant, female, 42-year-old).

**Recognition from the community.** CTC SRH workers maintain a good relationship with the community people including the gatekeepers and political leaders. Community people often praise them for the services they provide. They feel very proud when people praise their services. This keeps them going and inspires them to work harder for providing better SRH services. In this issue, one CTC SRH worker told that *"Their [community people's] affection and respect for me encourage me to provide better service...We must hear everything [they tell] with attention. If I pay attention to them, they will say that I am good and generous to them."* (Family welfare visitor, female, 43-year-old).

People living in these low-income and informal urban settlements mostly rely on the CTC SRH workers for their SRH needs, particularly for menstrual regulation. Hence, when any worker faces any problem during their field visit community people help them. It would not be possible for the providers to work in the field without the acceptance of the community people. In this regard, a CTC SRH worker mentioned, *"People in this area give me tremendous support. However, few people say that I refer people for my own sake. However, most people appreciate my services. I have good relationships with people. Good relations are helpful for my work too. Sometimes I must visit the community even at 3:00 a.m.. . .Serving people is my main concern."* (Community health volunteer, female, 30-year-old).

**Recognition from the clients.** We found that a sense of being respected and valued by the clients motivates the CTC SRH workers. Many clients in the informal urban settlements are vulnerable women in need of SRH services. They value the service and friendly behaviour of the SRH workers and, in return for the service, respect them, and feel happy to see them. For example, a female service promoter said, *"I like [working with] the pregnant women because when we visit them, they feel happy. They welcome us with a smile and say, 'sisters have come!'"*

**Clients' compliance.** Clients' compliance improves the motivation of the CTC SRH workers. Many SRH issues are highly stigmatized and hence take a lot of toll on the workers. Therefore, when they find that clients are following their instructions and visited the formal health facility after the referral, they feel rewarded. This is illustrated by the following quotation from a female service promoter: *"I feel good when the patients [clients] listen to us and take up the referral, and her husband follows our instructions too. . . It's inspiring when they listen to what we have told them when they come [taking up the referral] here [clinic] following our words. Even mothers-in-law bring them here! Just a few days ago a mother-in-law brought her daughter-in-law here for menstrual regulation. They came here after our counseling. This is a great inspiration for us."*

Referral uptake by their clients is very important to the CTC SRH workers because it has both monetary and non-monetary implications. This is evident in the following statement:

*"When the patients come, the recognition is good. If not, then we must hear [from the line manager] that "you are not doing fieldwork properly; not going to the field regularly. Therefore, the clients are not coming" . . . If I get recognition, then I get motivated to work better. [I work hard] thinking if I work better then I will get better appreciation. Sometimes, I feel disappointed because I'm working well, but if people don't come, I can't force them to come."* (Health educator, female, 27-year-old).

## Discussion

The role of close-to-community (CTC) sexual and reproductive health (SRH) workers is vital to promote universalism, equality, non-discrimination, and inclusion in SRH services. They are useful to ensure that the SRH needs and rights of people living in low-income urban settlements are met. Here we investigated the factors that motivate or demotivate CTC SRH workers to continue their job in low-income urban settlements in Bangladesh.

Our findings reveal a range of factors that affect the motivation of CTC SRH workers in poor urban settlements in Bangladesh. As Herzberg and colleagues suggested in their two-factor theory on work motivation [24], the motivation of CTC SRH workers in low-income urban settlements in Bangladesh is influenced by a range of extrinsic and intrinsic factors (Table 2). These factors are related to their work, community, clients, personal, career, work environment, monetary incentives, and supervision. Similar findings are also reported by studies conducted elsewhere [25–30]. Factors identified in our study are in line with the existing models of work motivation, such as Herzberg's two-factor models of work motivation [24,31] and the six primary needs or motives of the employees in a work setting proposed by Pareek [32].

Globally, CHWs mainly work voluntarily without or with very little financial incentive [21]. Our research participants included volunteer CTC workers and a mix of fully paid workers and workers paid for performance or commodity selling. Poor financial incentives, job insecurity, and lack of retirement benefits are identified as sources of dissatisfaction among both salaried and volunteer CTC SRH workers. It should be noted that CHWs mostly come from low-income backgrounds [21]. Giving time without or with limited financial benefit is a missed income opportunity for them. When there is a need for income or higher income, they might consider quitting their job and looking for alternative income-generating activities. The same is reported by many studies in Asia and Africa [33]. The commonly held belief is that paying CHWs is not sustainable [21]. But, when CHWs are in financial insecurity, they are more likely to quit and provide poor-quality service [33]. Recruiting new CHWs will be expensive because of the training and other resources need in the management [34]. Premature dropout of CHWs has a negative impact on the finances of the programme. Hence, this might also make a CHW programme unsustainable [34]. Our respondents unanimously expressed their frustration about the lack of or poor financial incentives and job insecurity. They believe that the amount of incentive they were receiving does not match their workload. In a country, where the unemployment rate is high, often people might take the job of CHWs to move up the career ladder or move to new employment. Herzberg noted that workers are more likely to be dissatisfied if they perceive unfairness in their financial incentives [24,31]. This dissatisfaction may work as a push factor [24,31] and may result in increased dropout of the CTC SRH workers. Therefore, we argue that CTC SRH workers should be adequately compensated for the time they devoted to the job.

Supportive, and regular, supervision was identified as another important extrinsic factor. More supportive approaches in supervision were mentioned as a positive factor in the literature as well [35]. Supervisors should act as mentors of the community SRH workers. They should monitor their performance and give appropriate feedback to increase the efficiency of their work [36]. Although supervision is considered an extrinsic factor, CTC SRH workers might consider it important to fulfill their career aspirations and growth needs. Hence, supportive, and regular supervision will positively affect the intrinsic factors/motivators of the CTC SRH workers.

A positive and trusting relationship between the health service providers, community, client, and the health system is important for the success of any health programme [37,38]. This

kind of relationship maintains satisfaction and prevents dissatisfaction among the CTC SRH workers. We found that CTC SRH workers understand the importance of these relationships and strive to maintain and improve their relationships with the other stakeholders of the community SRH services.

We argue that the physical safety and security of the CTC SRH workers are given adequate attention by the community health organizations. Law and order in low-income urban settlements in LMICs are not often promising, as also reported by our respondents. A recent systematic review on the influence of the context on the performance of the CHWs reported that CHWs, particularly women, are concerned with their safety and this safety concern affects their motivation, and performance and often leads to dropout [33]. Therefore, community health organizations should develop strategies to minimize the CTC SRH workers' physical risks in the community.

Along with the extrinsic or hygiene factors, a range of intrinsic factors or motivators were also found to drive the motivation of our respondents (Table 2). We found that CTC SRH workers find the motivation to continue their work when they believe or observe that their work has a positive impact on people's lives and that the services they are providing are of good quality. Observing the positive impact of work is also reported as a strong motivator for the CHWs in the literature [21,39]. CHWs have varying levels of education [40] ranging from no education to a university degree in non-health fields in this current study. Therefore, in addition to appropriate training and orientation at recruitment, regular refresher training is vital to equip them with the necessary knowledge and skills to carry out the job. Training opportunities might also work as an incentive for the CTC SRH workers since this will help them to fulfill their career and personal growth aspirations.

There is also a spiritual dimension of motivation- satisfaction in serving people and serving God by serving the community. CTC SRH workers are often from the same or a similar community [33] and spend a great deal of time in the community. Hence, they understand the vulnerability of their clients- mostly women with very low socio-economic status. They take a great deal of pride and satisfaction, consequently motivation to continue, knowing that they are serving a vulnerable group in great need of SRH services. This spiritual dimension of the motivating factor was also evident in studies conducted elsewhere. For example, a study in Tanzania found that a desire to serve the God motivated the CHWs to become CHWs [41].

One of the most cited factors of motivation by our respondents is career growth and development opportunities. These factors also include the opportunity to improve knowledge and skills, and professional networking. This has also been reported consistently in the literature [21]. Career stagnation is demotivating. CHWs want to move up the career ladder. A hope to get a promotion might work as fuel for hard work and good service, but a lack of such hope might do just the opposite. However, to get a promotion, along with sincere work and quality performance, they also need to develop their skills and knowledge. Hence, to keep CTC SRH workers motivated, in addition to promotion opportunities there should be training opportunities for the CTC SRH workers to improve their knowledge and skill in community SRH services. CHWs view gaining new knowledge and skills as a path to better employment or promotion opportunities in the same organization [21].

Our respondents unanimously identified recognition of their services from the client, community, and higher-level authority as an important determinant of their motivation. Evidence suggests that lack of support and respect from the higher-level authority and or the community negatively affect the motivation and performance of the CHWs [38].

One of the main jobs of the CTC SRH workers in low-income urban settlements in Bangladesh is to refer women with SRH service needs, including menstrual regulation, to the clinic they are affiliated with. Referral uptake by the clients is considered a key performance indicator

of the CTC SRH workers. We found that successful referrals to health facilities work as a strong motivator among CTC SRH workers in Bangladesh. CHW literature also indicates successful referral to a health facility as a complementary incentive to the CHWs [42]. However, CTC SRH workers in Bangladesh operate in a pluralistic setting, where there is stiff competition between different formal, informal, and private providers. The majority of our respondents reported difficulty in achieving the referral targets. Further research is needed to understand the factors affecting referral uptake by women with SRH needs in low-income urban settlements in Bangladesh.

It is important to keep CTC SRH workers motivated to prevent premature dropout. A high dropout is related to poor programme performance and higher programme costs. An important theory to understand what motivates an individual in a work condition is the Maslow's Hierarchy of Needs [43]. Extrinsic factors identified in our study lie in the bottom tiers of the Maslow's Hierarchy of Needs, whereas the motivators or intrinsic factors can be placed in the upper tiers. It is suggested that unless an individual's basic needs have been met, higher levels in the hierarchy are of no relevance [43,44]. A study on nurses in New Zealand also found that most cited concerns are related to hygiene factors [45]. Therefore, organizations need to make sure that these hygiene factors are adequately met. Unmet hygiene factors might demotivate the CTC SRH workers and cause them to work less hard.

Our studies included both salaried and volunteer CTC SRH workers and they were recruited from both the government and NGO sectors. However, we did not compare motivation between these groups. It would be interesting to know how these CTC SRH worker groups differ in terms of their work motivation. Further study should focus on that.

## Conclusion

A range of extrinsic/hygiene factors and intrinsic factors/motivators affect the motivation of CTC SRH workers in low-income urban settlements in Bangladesh. Commonly cited reasons for dissatisfaction among the CTC SRH workers are poor salaries or other financial incentives, lack of job security, and absence of supportive supervision. While, the motivators include witnessing the positive impact of work, promotion, developing new knowledge and skills, and successful referral among others. Community health programmes are vital in promoting universalism and equity in the health of the population. CHWs are at the heart of any community health programmes. To prevent premature dropout and ensure quality community SRH services employers of the CTC SRH workers should give adequate emphasis on meeting both hygiene factors and motivators.

## Acknowledgments

REACHOUT was a 5-year international research consortium aiming to generate knowledge to strengthen the performance of community health service providers of promotional, preventive, and curative primary health services in rural and urban areas in Africa and Asia.

## Author Contributions

**Conceptualization:** Ilias Mahmud, Malabika Sarker, Sally Theobald, Sabina Faiz Rashid.

**Data curation:** Sumona Siddiqua, Irin Akhter, Sabina Faiz Rashid.

**Formal analysis:** Ilias Mahmud, Sumona Siddiqua.

**Funding acquisition:** Malabika Sarker, Sabina Faiz Rashid.

**Investigation:** Ilias Mahmud, Malabika Sarker, Sabina Faiz Rashid.

**Methodology:** Ilias Mahmud, Malabika Sarker, Sally Theobald, Sabina Faiz Rashid.

**Project administration:** Ilias Mahmud, Irin Akhter, Malabika Sarker, Sabina Faiz Rashid.

**Resources:** Malabika Sarker, Sabina Faiz Rashid.

**Supervision:** Malabika Sarker, Sally Theobald, Sabina Faiz Rashid.

**Writing – original draft:** Ilias Mahmud, Sumona Siddiqua, Irin Akhter.

**Writing – review & editing:** Ilias Mahmud, Malabika Sarker, Sally Theobald, Sabina Faiz Rashid.

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
