## [Decision Letter · Decision Letter 0]

20 Dec 2021

PONE-D-21-35310Factors affecting motivation of close-to-community sexual and reproductive health workers in low-income urban settlements in Bangladesh: a qualitative studyPLOS ONE

Dear Dr. Mahmud,

Thank you for submitting your manuscript to PLOS ONE. After careful consideration, we feel that it has merit but does not fully meet PLOS ONE’s publication criteria as it currently stands. Therefore, we invite you to submit a revised version of the manuscript that addresses the points raised during the review process. Please see the comments from both the reviewers and addreess them in your revised manuscript. 

We look forward to receiving your revised manuscript.

Kind regards,

Alok Ranjan

Academic Editor

PLOS ONE

Journal Requirements:

Reviewers' comments:

Reviewer's Responses to Questions

**Comments to the Author**

1. Is the manuscript technically sound, and do the data support the conclusions?

Reviewer #1: Yes

Reviewer #2: Yes

2. Has the statistical analysis been performed appropriately and rigorously? 

Reviewer #1: N/A

Reviewer #2: N/A

3. Have the authors made all data underlying the findings in their manuscript fully available?

Reviewer #1: Yes

Reviewer #2: Yes

4. Is the manuscript presented in an intelligible fashion and written in standard English?

Reviewer #1: Yes

Reviewer #2: Yes

5. Review Comments to the Author

Reviewer #1: Please find below my comments for the manuscript. It is a good piece of work which can be strengthened with abstraction and more structure.

Methods:

1. Specifying how many participants were salaried and how many were volunteers in the study is important as it may/may not influence factors such as financial stability. Further, in the analysis, was there any difference in their motivation factors, especially for factors related to financial stability? Even if there wasn't any difference, it is important to specify that in the results and discuss what that implies. Please incorporate these in the manuscript.

2. Kindly make the following changes in Table 1:

A) Adding a column - salaried/volunteer would help contextualize the findings of the paper better.

B) Also, adding a note below the table to provide some information regarding what are Marie Stopes volunteers and their major focus of work would be useful.

3. Some description of the interview guide describing the broader themes of questions, specific to the research question of the paper and not about the larger project, would be useful. Kindly include that.

4. While describing the analysis, using the framework technique, the authors state "This technique involved identifying, abstracting, charting, and matching themes (in our case, factors affecting motivation of CTC SRH workers) across the interviews." This description is vague and does not offer any clarity on how the analysis was performed in this paper. Please describe how the framework technique was used specifically for this paper in 2-4 sentences as it would be of greater value.

5. The authors use Herzberg's two-factor motivation theory to organize the findings of the paper. However, this is not explicitly stated in the Methods section (The theory is indeed mentioned in the discussion without any elaboration). This leads to enormous confusion while reading the manuscript. For eg: A) Intrinsic and Extrinsic factors could be based on multiple axis i.e. factors intrinsic/extrinsic to the health systems or factors intrinsic/extrinsic to individual CHWs. B) Why extrinsic factors are also called hygiene factors and why are intrinsic factors called motivators. And this confusion persists across the results section.

Please describe the Herzberg two-factor theory briefly in the methods section, specifying how intrinsic and extrinsic factors are conceptualized in the theory and how it has been used to organize the findings of the study as it would strengthen the paper tremendously.

Results:

6. The analysis is a bit weak currently as it lacks abstraction and proper structure. Most, if not all, subheadings in the Results section are codes with quotes. These codes need further synthesis, linking multiple codes together to describe broader ideas. This would reduce the number of subheadings in the result section making it more coherent and comprehensible. For example:

A) Constructs like “community attitudes towards CTC SRH workers”, “relationship between CTC workers and community” and “physical safety and security” are primarily community factors, whereas all the other constructs within extrinsic factors are factors within the health systems These community factors could be discussed together within one sub-heading and their narratives could be connected.

B) Constructs like “fair remuneration”, “job security”, and “retirement benefits” and closely linked to financial stability and could be described under one subheading.

C) Similarly, in Intrinsic Factors, constructs like “Impact of the work on clients and own lives” and “opportunity to serve vulnerable women” discuss the idea of how CTC SRH workers find value in their work.

D) Constructs like “opportunity to improve knowledge and skills” and “promotion opportunities” are about professional growth.

E) Constructs “professional networking”, “recognition from community, clients, and supervisors”, and “clients compliance” are all discussing some form of recognition and could be discussed together.

Overall, the results section needs more synthesis to make it crisp and the ideas could be organized to develop linkages across subheadings to ensure better flow.

7. Please modify Table 2 to provide 1) Theme i.e. Intrinsic and extrinsic, 2) subthemes (following the suggestions in comment 5, and 3) codes (that are enumerated in the table and the ones which appear in the text but are missing from the table) organized within appropriate subheadings. This would help the reader get a quick snapshot of the findings of the paper.

A) In Table 2: the construct “monitoring” is vague. It needs elaboration like the construct “supportive and regular supervision”.

B) There are inconsistencies between the headings in the text and the codes in the table. For example: "Challenges in meeting target in a pluralistic competitive environment" - This subheading appears in written text but finds no mention in Table 2; Monitoring appears as a separate construct in the table but is merged in the supervision section in the text. Please avoid such inconsistencies.

C) Headings like “fair remuneration”, “job security”, “retirement benefits” primarily discuss the ideas of unfair remuneration, lack of job security, and lack of retirement benefits. Reading the table implies that workers get fair remuneration, whereas the text implies the exact opposite. Please modify this to avoid confusion.

8. Subheading - Recognition from Supervisors: Both the quotes "Supervisor compliments us if he likes our work. Then, he advises us on how to work better by improving our skills.” and “We have the freedom. We work the way we want to. I do every work in my own responsibility” and indicative of supportive supervision but they are not indicative of the code “Recognition from supervisor” as a motivator for their work. Please add quotes reflective of the code.

9. Subheading - Impact of work on clients and their own lives: The quotes have little to do with the impact of their work. They are indicative of how the community health workers are developing a sense of satisfaction and finding value in the work they do. This subheading needs to be changed.

10. Subheading - Challenges in meeting target in a pluralistic competitive environment: All quotes in this section are indicative of pressure from superiors to meet targets, and non-recognition of the work CTC SRH workers do. However, the heading is not indicative of this at all. The quotes do not discuss any specific challenges to achieving targets, they just indicate not being able to achieve targets. No quotes indicate the role of a pluralistic competitive environment as a challenge.

Discussion:

11. Please check the formatting, there are inconsistencies in the font.

12. The following statement is confusing. "Factors identified in our study confirm the existing models of work motivation, such as Herzberg’s two-factor models of work motivation." Was the purpose of this research to test Herzberg's two-factor theory or did the authors use the two-factor theory to organize and make sense of the findings? Based on this draft, the authors seem to have done the latter, but the statement claims otherwise.

13. "Poor financial incentive, job insecurity, and lack of retirement benefit are identified as sources of dissatisfaction among our participants. It should be noted that CHWs mostly come from a low-income background." While discussing this, it would be important to discuss the findings based on comment 1 about the difference or lack of it between salaried and volunteering employees.

14. "But, when CHWs are in financial insecurity, they are more likely to quit and provide poor quality service." Please support this claim with appropriate references.

15. "Since often CHWs works with none or inadequate financial incentive, it is of prime importance that they receive the due recognition of the invaluable services they are rendering in low-income communities." This statement implies that they should be recognized primarily because they are receiving low financial incentives, which is not the case. Recognition for work is independent of the financial incentive. This statement pits these two constructs against each other i.e. either pay a good salary or recognize them for their work. Kindly edit this.

16. "An important theory to understand what motivates an individual in a work condition is the Maslow's Hierarchy of Needs [43]. Extrinsic factors identified in our study lie in the bottom tiers of the Maslow's Hierarchy of Needs, whereas the motivators or the intrinsic factors can be placed in the upper tiers." This is a relevant theory to weave all the findings together and contextualize the phenomenon in a broader sense. However, it is not elaborated enough and the paragraph ends rather abruptly. Considering this is the last paragraph of the Discussion section, it would be better to elaborate this idea using the findings of this study (briefly summarizing the findings based on where they lie in Maslow's hierarchy) and specifying the health systems implications (enumerating some tangible strategies).

17. The structure of discussion would need some modification after re-organizing the results to ensure a better flow of ideas.

All the Best to the authors. Look forward to reading the paper.

Reviewer #2: Factors affecting motivation of close-to-community sexual and reproductive health workers in low-income urban settlements in Bangladesh: a qualitative study

Thank you for an opportunity to review this paper, and my hearty congratulations on putting together a paper on an important topic. It is indeed important to bring voices of CTC providers to the forefront, and papers like this add to much needed literature from lower- and middle-income countries and led by local authors.

I have some suggestions on the paper, which I hope will help to strengthen it. My apologies for the long comments. The paper does need a fair amount of rework for an external audience that is not familiar with the Bangladesh context to understand it better. Some section-wise comments are below:

Abstract

• In the abstract CTC health workers and CHWs are used interchangeably. It is a bit confusing at this point to understand whether the paper is about frontline providers (doctors, nurses and outreach) or on CHWs only.

• There have been several papers on the motivation of health workers that churn up a whole list of factors. Perhaps it might help to say what is unique about this paper and what its findings add to what already exists. Perhaps none from Bangladesh? or maybe few papers have used the framework you have used? etc

• I like the conclusion sentence.

• I don’t see the use of the framework technique of analyses (case versus themes) in the findings.

Background

• The context of CTC/CHW in Bangladesh needs to be explained for an external audience. Are all the CTC providers in this study are from NGOs? I referred to another paper referred by the authors (reference 22, Mahmud et al. 2015) which has a good description of the context. It might help to rewrite the first bit.

• It would be nice to have some sense of how the CTC providers are paid, how they are recruited, are they contractual workers -perhaps a summary table. since all this gets referred to the factors influencing their motivation later but we don’t get a context here. Are they given incentives, paid salaries, etc?

• Perhaps it might help to say what is unique about this paper and what its findings add to general literature on CTC providers’ motivation. (What do we already know from literature, and what are we adding here)

Conceptual/analytical framework used.

• It would help to have an explanation on where the concept of intrinsic and extrinsic factors is coming from. Without some background to the framework or a reference to it, it might be difficult for readers to understand why a factor is classified as extrinsic or intrinsic. Is table 2 the framework? Has it been adapted? (or is table 2 specific to Bangladesh findings?)

• Why was this framework used rather than other similar frameworks on motivation for the analysis?

• From my understanding of the conceptual framework, the intrinsic factors are round recognition, fairness, confidence building, learning in the job, appreciation, etc (not quality of SRH or compliance). It might help to reword the headings of this set of factors a little so as to make these factors sound more intrinsic.

Methods

• Some details of how the analysis was done would be helpful.

Results

• There are rich, interesting points in the findings section.

• I am interested in knowing the difference between extrinsic and intrinsic factors. For instance, why is community attitudes an extrinsic factor and client compliance an intrinsic factor? These sound like related themes.

• Table 2 headings and the headings of the section themes do not always match. There is also some repetition between extrinsic and intrinsic factors descriptions.

Page 7- Community attitudes and relationship with the community

We found that community mostly have positive attitudes towards CTC SRH workers and their services

We probably can’t make this statement without interviewing the community. needs to be reworded.

Page 10- Fair remuneration- It is tough to understand without a context of how CTCs were renumerated in the first place. Was it incentives? Salaries?

• It might help to follow the standard way in which quotes from participants are put in qualitative papers. Referring to previous PLOS one papers for this format might help. Also, all quotes need a participant reference (some have been missed right now)

• While all the factors are stated and described, how these factors affect motivation of CHWs is often not described. I feel that if the authors can tease out these ‘how’ mechanisms of influence much better, and that could be the strength of this paper.

-For instance, in the community attitudes section, the attitudes of the community are described. But how these attitudes link to CHW motivation is not mentioned.

-Another instance, if we say supportive supervision is important, what are the mechanisms through which it is affecting motivation- is it through learning and mentorship , confidence building, general presence of supervisors, etc. How is poor supervision influencing motivation adversely? These are all there in the description, but I think it would be interesting for readers if the ideas in this section are sorted better.

• Right now, the findings read like a big list of factors with little interconnectedness. One can point out stories of connection- how a whole range of factors influence motivation. Which is why the policy interventions also need to be bundled.

• It is also hard to make out which were the most important factors and what were secondary and mentioned only by few respondents. In a framework analysis, this should be possible.

Discussion

• It might help to re-frame the discussion a little- in the light of what is unique about this study and what the study contributed to literature. In my mind, many studies have pointed out factors that affect motivation, but few have tried to elucidated mechanisms through which these factors work. A lot of these mechanisms are in the findings but a bit jumbled. It might help to highlight these. Also, few studies have used a framework like this to cluster the factors, etc.

• What factors were similar to what has been found in other studies. Also, what factors were different. What was different perhaps in the way a few factors played out in comparison to other studies. How does this study contribute to existing literature on CHWs? Were all factors equally important? What factors played out positively in your setting, and what were deterrents?

• Throughout the discussion, CHWS and CTC providers are used interchangeably.

• Page 17- Factors identified in our study confirm the existing models of work motivation, such as the Herzberg’s two-factor models of work motivation [24, 31] and the six primary needs or motives of the employees in a work setting proposed by Pareek [32]. Perhaps this has to come earlier. There is no explanation about what the 6 primary needs are- was this the basis for the analysis or just another framework that was considered. How were the frameworks adapted to the results of this study

• Page 18: The commonly held believe is that paying CHWs is not sustainable [21]- Sentence needs to be nuanced. In India for example, CHWs like ASHAs are not paid fixed salaries, but they do get incentives.

• Page 18 Premature dropout of CHWs has negative impact on the finance of the programme. Hence, this will make a CHW programme unsustainable [34]. Needs to be toned down.

• Page 20- Since often CHWs works with none or inadequate financial incentive, it is of prime importance that they receive the due recognition of the invaluable services they are rendering in low-income communities. Needs to be toned or rephrased for it sounds as though we are okay with CHWs being paid less

• A section on the limitations of the study would also be of value.

Once again, thank you for the opportunity to review this paper. I think the paper has a lot of value but needs some more work so as to help readers understand the research better. My very best wishes.

6. PLOS authors have the option to publish the peer review history of their article (what does this mean?). If published, this will include your full peer review and any attached files.

Reviewer #1: No

Reviewer #2: No

---

## [Author Response · Author response to Decision Letter 0]

29 Jun 2022

Response to review comments

1. Is the manuscript technically sound, and do the data support the conclusions?

Reviewer #1: Yes

Reviewer #2: Yes

2. Has the statistical analysis been performed appropriately and rigorously?

Reviewer #1: N/A

Reviewer #2: N/A

3. Have the authors made all data underlying the findings in their manuscript fully available?

Reviewer #1: Yes

Reviewer #2: Yes

4. Is the manuscript presented in an intelligible fashion and written in standard English?

Reviewer #1: Yes

Reviewer #2: Yes

Reviewer #1: Please find below my comments for the manuscript. It is a good piece of work which can be strengthened with abstraction and more structure.

Response: We highly appreciate your valuable comments. Please find below our responses to your kind comments.

Methods:

1. Specifying how many participants were salaried and how many were volunteers in the study is important as it may/may not influence factors such as financial stability. Further, in the analysis, was there any difference in their motivation factors, especially for factors related to financial stability? Even if there wasn't any difference, it is important to specify that in the results and discuss what that implies. Please incorporate these in the manuscript.

Response: We have clarified this in the methods section including in Table 1 (added a column on job status). We have also incorporated this in our analysis and discussion. Please see the manuscript version tracking all the changes.

2. Kindly make the following changes in Table 1:

A) Adding a column - salaried/volunteer would help contextualize the findings of the paper better.

Response: Thanks! We have added the suggested column. 

B) Also, adding a note below the table to provide some information regarding what are Marie Stopes volunteers and their major focus of work would be useful.

Response: We have added a column in the table to provide information on the job for all the positions including Marie Stopes volunteers

3. Some description of the interview guide describing the broader themes of questions, specific to the research question of the paper and not about the larger project, would be useful. Kindly include that.

Response: We mentioned ‘First phase interviews were intended to understand the overall challenges of the CTC SRH workers including their motivation. While the second phase interviews exclusively explored the factors affecting their motivation.’ We hope this is enough.

4. While describing the analysis, using the framework technique, the authors state "This technique involved identifying, abstracting, charting, and matching themes (in our case, factors affecting motivation of CTC SRH workers) across the interviews." This description is vague and does not offer any clarity on how the analysis was performed in this paper. Please describe how the framework technique was used specifically for this paper in 2-4 sentences as it would be of greater value.

Response: We have further clarified analysis. Please see the last paragraph of the revised method section.

5. The authors use Herzberg's two-factor motivation theory to organize the findings of the paper. However, this is not explicitly stated in the Methods section (The theory is indeed mentioned in the discussion without any elaboration). This leads to enormous confusion while reading the manuscript. For eg: A) Intrinsic and Extrinsic factors could be based on multiple axis i.e. factors intrinsic/extrinsic to the health systems or factors intrinsic/extrinsic to individual CHWs. B) Why extrinsic factors are also called hygiene factors and why are intrinsic factors called motivators. And this confusion persists across the results section.

Please describe the Herzberg two-factor theory briefly in the methods section, specifying how intrinsic and extrinsic factors are conceptualized in the theory and how it has been used to organize the findings of the study as it would strengthen the paper tremendously.

Response: Thanks for pointing this out. We have clarified this in the method section where we discussed analysis. Please see the last paragraph of the method section. 

Results:

6. The analysis is a bit weak currently as it lacks abstraction and proper structure. Most, if not all, subheadings in the Results section are codes with quotes. These codes need further synthesis, linking multiple codes together to describe broader ideas. This would reduce the number of subheadings in the result section making it more coherent and comprehensible. For example:

A) Constructs like “community attitudes towards CTC SRH workers”, “relationship between CTC workers and community” and “physical safety and security” are primarily community factors, whereas all the other constructs within extrinsic factors are factors within the health systems These community factors could be discussed together within one sub-heading and their narratives could be connected.

Response: revised accordingly

B) Constructs like “fair remuneration”, “job security”, and “retirement benefits” and closely linked to financial stability and could be described under one subheading.

Response: Revised accordingly.

C) Similarly, in Intrinsic Factors, constructs like “Impact of the work on clients and own lives” and “opportunity to serve vulnerable women” discuss the idea of how CTC SRH workers find value in their work.

Response: Revised accordingly.

D) Constructs like “opportunity to improve knowledge and skills” and “promotion opportunities” are about professional growth.

Response: Revised accordingly.

E) Constructs “professional networking”, “recognition from community, clients, and supervisors”, and “clients compliance” are all discussing some form of recognition and could be discussed together.

Response: Revised accordingly.

Overall, the results section needs more synthesis to make it crisp and the ideas could be organized to develop linkages across subheadings to ensure better flow.

Response: Thanks! We consider these as excellent comments, and we revised our manuscript accordingly.

7. Please modify Table 2 to provide 1) Theme i.e. Intrinsic and extrinsic, 2) subthemes (following the suggestions in comment 5, and 3) codes (that are enumerated in the table and the ones which appear in the text but are missing from the table) organized within appropriate subheadings. This would help the reader get a quick snapshot of the findings of the paper.

Response: Revised accordingly.

A) In Table 2: the construct “monitoring” is vague. It needs elaboration like the construct “supportive and regular supervision”.

Response: We deleted monitoring. It was discussed under supervision as an example of unsupportive supervision.

B) There are inconsistencies between the headings in the text and the codes in the table. For example: "Challenges in meeting target in a pluralistic competitive environment" - This subheading appears in written text but finds no mention in Table 2; Monitoring appears as a separate construct in the table but is merged in the supervision section in the text. Please avoid such inconsistencies.

Response: We deleted monitoring from Table 2. It was discussed under supervision as an example of unsupportive supervision.

C) Headings like “fair remuneration”, “job security”, “retirement benefits” primarily discuss the ideas of unfair remuneration, lack of job security, and lack of retirement benefits. Reading the table implies that workers get fair remuneration, whereas the text implies the exact opposite. Please modify this to avoid confusion.

Response: In tables these are presented as factors which affected motivation of the CTC workers. Among our respondents some of these were positively present and some negatively. We tried to use neutral language in Tables and headings. For example, supportive and regular supervision, were found present in some CTC workers while absent in others. Therefore, in revised Table 2 and headings we tried to use neutral language such. We discussed our findings under the respective headings. 

8. Subheading - Recognition from Supervisors: Both the quotes "Supervisor compliments us if he likes our work. Then, he advises us on how to work better by improving our skills.” and “We have the freedom. We work the way we want to. I do every work in my own responsibility” and indicative of supportive supervision but they are not indicative of the code “Recognition from supervisor” as a motivator for their work. Please add quotes reflective of the code.

Response: We agree, this is a cross-cutting quotes. We believe it can also be applied for recognition. This is the best quote we have on recognition.

9. Subheading - Impact of work on clients and their own lives: The quotes have little to do with the impact of their work. They are indicative of how the community health workers are developing a sense of satisfaction and finding value in the work they do. This subheading needs to be changed.

Response: we rearranged this section under ‘perceived value in the work’ sub-heading.

10. Subheading - Challenges in meeting target in a pluralistic competitive environment: All quotes in this section are indicative of pressure from superiors to meet targets, and non-recognition of the work CTC SRH workers do. However, the heading is not indicative of this at all. The quotes do not discuss any specific challenges to achieving targets, they just indicate not being able to achieve targets. No quotes indicate the role of a pluralistic competitive environment as a challenge.

Response: we rearranged this section and discussed this under workload/targets.

Discussion:

11. Please check the formatting, there are inconsistencies in the font.

Response: corrected

12. The following statement is confusing. "Factors identified in our study confirm the existing models of work motivation, such as Herzberg’s two-factor models of work motivation." Was the purpose of this research to test Herzberg's two-factor theory or did the authors use the two-factor theory to organize and make sense of the findings? Based on this draft, the authors seem to have done the latter, but the statement claims otherwise.

Response: Yes, you are right. We have revised this section.

13. "Poor financial incentive, job insecurity, and lack of retirement benefit are identified as sources of dissatisfaction among our participants. It should be noted that CHWs mostly come from a low-income background." While discussing this, it would be important to discuss the findings based on comment 1 about the difference or lack of it between salaried and volunteering employees.

Response: We clarified this further.

14. "But, when CHWs are in financial insecurity, they are more likely to quit and provide poor quality service." Please support this claim with appropriate references.

Response: Done

15. "Since often CHWs works with none or inadequate financial incentive, it is of prime importance that they receive the due recognition of the invaluable services they are rendering in low-income communities." This statement implies that they should be recognized primarily because they are receiving low financial incentives, which is not the case. Recognition for work is independent of the financial incentive. This statement pits these two constructs against each other i.e. either pay a good salary or recognize them for their work. Kindly edit this.

Response: Thanks! Edited accordingly.

16. "An important theory to understand what motivates an individual in a work condition is the Maslow's Hierarchy of Needs [43]. Extrinsic factors identified in our study lie in the bottom tiers of the Maslow's Hierarchy of Needs, whereas the motivators or the intrinsic factors can be placed in the upper tiers." This is a relevant theory to weave all the findings together and contextualize the phenomenon in a broader sense. However, it is not elaborated enough and the paragraph ends rather abruptly. Considering this is the last paragraph of the Discussion section, it would be better to elaborate this idea using the findings of this study (briefly summarizing the findings based on where they lie in Maslow's hierarchy) and specifying the health systems implications (enumerating some tangible strategies).

Response: added a couple of sentences to clarify this further.

17. The structure of discussion would need some modification after re-organizing the results to ensure a better flow of ideas.

Response: We have organized discussion following the revised results section.

All the Best to the authors. Look forward to reading the paper.

Reviewer #2: Factors affecting motivation of close-to-community sexual and reproductive health workers in low-income urban settlements in Bangladesh: a qualitative study

Thank you for an opportunity to review this paper, and my hearty congratulations on putting together a paper on an important topic. It is indeed important to bring voices of CTC providers to the forefront, and papers like this add to much needed literature from lower- and middle-income countries and led by local authors.

I have some suggestions on the paper, which I hope will help to strengthen it. My apologies for the long comments. The paper does need a fair amount of rework for an external audience that is not familiar with the Bangladesh context to understand it better. Some section-wise comments are below:

Abstract

• In the abstract CTC health workers and CHWs are used interchangeably. It is a bit confusing at this point to understand whether the paper is about frontline providers (doctors, nurses and outreach) or on CHWs only.

Response: we have updated and maintained CTC health workers throughout the abstract. They are not professionals but included both volunteer and salaried CTC. Further details are provided in methods section.

• There have been several papers on the motivation of health workers that churn up a whole list of factors. Perhaps it might help to say what is unique about this paper and what its findings add to what already exists. Perhaps none from Bangladesh? or maybe few papers have used the framework you have used? Etc

Response: most work was done on volunteer CTC workers, but we included both volunteer and salaried workers.

• I like the conclusion sentence.

Response: Thanks!

• I don’t see the use of the framework technique of analyses (case versus themes) in the findings.

Response: We used the Herzberg’s two-factor models of work motivation to fit our findings in.

Background

• The context of CTC/CHW in Bangladesh needs to be explained for an external audience. Are all the CTC providers in this study are from NGOs? I referred to another paper referred by the authors (reference 22, Mahmud et al. 2015) which has a good description of the context. It might help to rewrite the first bit.

Response: CTC providers in this study are from both NGOs and government sectors. We clarified this in methods section. In introduction section, we mentioned that these services are mainly provided by the NGOs.

• It would be nice to have some sense of how the CTC providers are paid, how they are recruited, are they contractual workers -perhaps a summary table. since all this gets referred to the factors influencing their motivation later but we don’t get a context here. Are they given incentives, paid salaries, etc?

Response: We have added a column in Table 1 to clarify their salaried or volunteer status.

• Perhaps it might help to say what is unique about this paper and what its findings add to general literature on CTC providers’ motivation. (What do we already know from literature, and what are we adding here)

Response: We have briefly addressed this in the last two paragraphs of the background section.

Conceptual/analytical framework used.

• It would help to have an explanation on where the concept of intrinsic and extrinsic factors is coming from. Without some background to the framework or a reference to it, it might be difficult for readers to understand why a factor is classified as extrinsic or intrinsic. Is table 2 the framework? Has it been adapted? (or is table 2 specific to Bangladesh findings?)

Response: We have clarified this further in the last paragraph of the methods section.

• Why was this framework used rather than other similar frameworks on motivation for the analysis?

Response: This framework is well known to understand work motivation. However, in discussion we have also compared our findings against the model proposed by Pareek and the Maslow's Hierarchy of Needs.

• From my understanding of the conceptual framework, the intrinsic factors are round recognition, fairness, confidence building, learning in the job, appreciation, etc (not quality of SRH or compliance). It might help to reword the headings of this set of factors a little so as to make these factors sound more intrinsic.

Response: We have reorganized the whole results section and Table 2.

Methods

• Some details of how the analysis was done would be helpful.

Response: We have further clarified analysis in methods section.

Results

• There are rich, interesting points in the findings section.

• I am interested in knowing the difference between extrinsic and intrinsic factors. For instance, why is community attitudes an extrinsic factor and client compliance an intrinsic factor? These sound like related themes.

• Table 2 headings and the headings of the section themes do not always match. There is also some repetition between extrinsic and intrinsic factors descriptions.

Page 7- Community attitudes and relationship with the community

We found that community mostly have positive attitudes towards CTC SRH workers and their services

We probably can’t make this statement without interviewing the community. needs to be reworded.

Page 10- Fair remuneration- It is tough to understand without a context of how CTCs were renumerated in the first place. Was it incentives? Salaries?

• It might help to follow the standard way in which quotes from participants are put in qualitative papers. Referring to previous PLOS one papers for this format might help. Also, all quotes need a participant reference (some have been missed right now)

• While all the factors are stated and described, how these factors affect motivation of CHWs is often not described. I feel that if the authors can tease out these ‘how’ mechanisms of influence much better, and that could be the strength of this paper.

-For instance, in the community attitudes section, the attitudes of the community are described. But how these attitudes link to CHW motivation is not mentioned.

-Another instance, if we say supportive supervision is important, what are the mechanisms through which it is affecting motivation- is it through learning and mentorship , confidence building, general presence of supervisors, etc. How is poor supervision influencing motivation adversely? These are all there in the description, but I think it would be interesting for readers if the ideas in this section are sorted better.

• Right now, the findings read like a big list of factors with little interconnectedness. One can point out stories of connection- how a whole range of factors influence motivation. Which is why the policy interventions also need to be bundled.

• It is also hard to make out which were the most important factors and what were secondary and mentioned only by few respondents. In a framework analysis, this should be possible.

Response: We have reorganized the whole results section. We have included both salaried and volunteer CTC workers. We further clarified this in the methods section. And in table 1. Table one included two new columns- one for salary status and another for job description. 

Regarding community attitude, we mentioned … as perceived/reported by the CTC workers.

We have regrouped findings under new headings and sub-headings. These are also incorporated in Table 2.

Discussion

• It might help to re-frame the discussion a little- in the light of what is unique about this study and what the study contributed to literature. In my mind, many studies have pointed out factors that affect motivation, but few have tried to elucidated mechanisms through which these factors work. A lot of these mechanisms are in the findings but a bit jumbled. It might help to highlight these. Also, few studies have used a framework like this to cluster the factors, etc.

• What factors were similar to what has been found in other studies. Also, what factors were different. What was different perhaps in the way a few factors played out in comparison to other studies. How does this study contribute to existing literature on CHWs? Were all factors equally important? What factors played out positively in your setting, and what were deterrents?

Response: in revised manuscript we discussed our findings in relation to existing literature. 

• Throughout the discussion, CHWS and CTC providers are used interchangeably.

Response: in revised manuscript we have used CTC providers consistently.

• Page 17- Factors identified in our study confirm the existing models of work motivation, such as the Herzberg’s two-factor models of work motivation [24, 31] and the six primary needs or motives of the employees in a work setting proposed by Pareek [32]. Perhaps this has to come earlier. There is no explanation about what the 6 primary needs are- was this the basis for the analysis or just another framework that was considered. How were the frameworks adapted to the results of this study

Response: in methods section we mentioned that we used the Herzber’s model of work motivation to frame our findings but in discussion we also compared with other models.

• Page 18: The commonly held believe is that paying CHWs is not sustainable [21]- Sentence needs to be nuanced. In India for example, CHWs like ASHAs are not paid fixed salaries, but they do get incentives.

Response: CTC providers in most countries get some incentives but that is very negligible amount. Our participants included both volunteer CTC who receive incentive and salaried CTC who receive a low salary.

• Page 18 Premature dropout of CHWs has negative impact on the finance of the programme. Hence, this will make a CHW programme unsustainable [34]. Needs to be toned down.

Response: edited

• Page 20- Since often CHWs works with none or inadequate financial incentive, it is of prime importance that they receive the due recognition of the invaluable services they are rendering in low-income communities. Needs to be toned or rephrased for it sounds as though we are okay with CHWs being paid less

Response: we deleted this sentence.

• A section on the limitations of the study would also be of value.

Response: We have added few sentences on limitations.

Once again, thank you for the opportunity to review this paper. I think the paper has a lot of value but needs some more work so as to help readers understand the research better. My very best wishes.

Response: Thanks for your kind notes and invaluable comments. We believe our manuscript greatly benefited from your comments.

---

## [Decision Letter · Decision Letter 1]

12 Sep 2022

PONE-D-21-35310R1Factors affecting motivation of close-to-community sexual and reproductive health workers in low-income urban settlements in Bangladesh: a qualitative studyPLOS ONE

Dear Dr. Mahmud,

Thank you for submitting your manuscript to PLOS ONE. After careful consideration, we feel that it has merit but does not fully meet PLOS ONE’s publication criteria as it currently stands. Therefore, we invite you to submit a revised version of the manuscript that addresses the points raised during the review process.

Please see the reviewer's comments and submit the revised manuscript.

We look forward to receiving your revised manuscript.

Kind regards,

Alok Ranjan

Academic Editor

PLOS ONE

Journal Requirements:

Reviewers' comments:

Reviewer's Responses to Questions

**Comments to the Author**

1. If the authors have adequately addressed your comments raised in a previous round of review and you feel that this manuscript is now acceptable for publication, you may indicate that here to bypass the “Comments to the Author” section, enter your conflict of interest statement in the “Confidential to Editor” section, and submit your "Accept" recommendation.

Reviewer #2: All comments have been addressed

2. Is the manuscript technically sound, and do the data support the conclusions?

Reviewer #2: Yes

3. Has the statistical analysis been performed appropriately and rigorously? 

Reviewer #2: Yes

4. Have the authors made all data underlying the findings in their manuscript fully available?

Reviewer #2: Yes

5. Is the manuscript presented in an intelligible fashion and written in standard English?

Reviewer #2: Yes

6. Review Comments to the Author

Reviewer #2: Thank you for sharing the great revisions in this paper which make it much easier for an external reader. I have a few minor points. 1. It might be easier to stick to one of the terms CTC health services provider or CTC health worker since both terms get used in the paper 2. Abstract- I would suggestto revise the sentence on doing analysis - one can say that the findings were analysed using pre-existing frameworks on motivation. 3. Table 2 themes and findings that follow from the table still dont completely seem to match,and might need minor edits 4. the section on organizational factors still feels a little long-winding, particularly the supportive supervison theme. One way to make the supportive supervison theme more readable is to analyse and link the paragraphs. for example -we can say that the data showed that supportive supervision helped in three ways- first , by improving skills. second, supervisors sometimes served as role-models and three, as a tracking mechanism. 5. The theme perceived value in the work is also a little long winding and feels repetitive, a little reorganising could make it read better

6. The paper might need one re-read from all authors to spot minor errors. for example the theme 'perceived impact of work on clients' starts with saying "We found that perceived positive impact of their work on ..."- and does not specify whom we were referring to as 'their' . Some tenses are also jumbledup after, which does happen when we write and rewrite- but if someone external can review once, it might be helpful.

thank you again and looking forward to seeing the paper.

7. PLOS authors have the option to publish the peer review history of their article (what does this mean?). If published, this will include your full peer review and any attached files.

Reviewer #2: No

---

## [Author Response · Author response to Decision Letter 1]

28 Oct 2022

Dear Editor,

We highly appreciate your and the reviewer’s invaluable contribution to improving our manuscript. We have addressed all editorial and peer review comments. Please find below our point-by-point response to all editorial and peer review comments.

Kind Regards,

Dr. Ilias Mahmud (on behalf of all authors)

Editor’s comment: 

Response: we have uploaded a rebuttal lettering providing a point-by-point response to all peer review comments.

Response: we have uploaded a revised manuscript with track changes. 

Response: We have uploaded an unmarked version of the revised manuscript incorporating all peer review comments.

Journal Requirements:

Response: We confirm that the reference list is complete and correct. We did not make any changes to the reference list during this revision. 

Reviewers' comments:

Comments to the Author

1. If the authors have adequately addressed your comments raised in a previous round of review and you feel that this manuscript is now acceptable for publication, you may indicate that here to bypass the “Comments to the Author” section, enter your conflict of interest statement in the “Confidential to Editor” section, and submit your "Accept" recommendation.

Reviewer #2: All comments have been addressed

Response: Thanks!

2. Is the manuscript technically sound, and do the data support the conclusions?

Reviewer #2: Yes

Response: Thanks!

3. Has the statistical analysis been performed appropriately and rigorously?

Reviewer #2: Yes

Response: Thanks!

4. Have the authors made all data underlying the findings in their manuscript fully available?

Reviewer #2: Yes

Response: Thanks!

5. Is the manuscript presented in an intelligible fashion and written in standard English?

Reviewer #2: Yes

Response: Thanks!

6. Review Comments to the Author

Reviewer #2: Thank you for sharing the great revisions in this paper which make it much easier for an external reader. I have a few minor points. 

Response: We highly appreciate your comments which we believe greatly benefited our manuscript. Please see below or point-by-point response to your specific comments.

1. It might be easier to stick to one of the terms CTC health services provider or CTC health worker since both terms get used in the paper 

Response: Thanks! We used CTC health workers throughout the paper.

2. Abstract- I would suggestto revise the sentence on doing analysis - one can say that the findings were analysed using pre-existing frameworks on motivation. 

Response: Thanks! We clarified this further in the abstract.

3. Table 2 themes and findings that follow from the table still dont completely seem to match,and might need minor edits 

Response: All the factors mentioned in Table 2 are discussed. We made minor adjustment/edits to improve clarity.

4. the section on organizational factors still feels a little long-winding, particularly the supportive supervison theme. One way to make the supportive supervison theme more readable is to analyse and link the paragraphs. for example -we can say that the data showed that supportive supervision helped in three ways- first , by improving skills. second, supervisors sometimes served as role-models and three, as a tracking mechanism. 

Response: Thanks! We have accepted and incorporated your suggestion.

5. The theme perceived value in the work is also a little long winding and feels repetitive, a little reorganising could make it read better

Response: We added an introductory paragraph to better link the information provided in the subsequent paragraphs. Also, we reorganized the paragraphs to improve the flow of information.

6. The paper might need one re-read from all authors to spot minor errors. for example the theme 'perceived impact of work on clients' starts with saying "We found that perceived positive impact of their work on ..."- and does not specify whom we were referring to as 'their' . Some tenses are also jumbledup after, which does happen when we write and rewrite- but if someone external can review once, it might be helpful.

thank you again and looking forward to seeing the paper.

Response: Thanks! We have edited the whole manuscript.

---

## [Editor Report · Decision Letter 2]

1 Dec 2022

Factors affecting motivation of close-to-community sexual and reproductive health workers in low-income urban settlements in Bangladesh: a qualitative study

PONE-D-21-35310R2

Dear Dr. Mahmud,

We’re pleased to inform you that your manuscript has been judged scientifically suitable for publication and will be formally accepted for publication once it meets all outstanding technical requirements.

Kind regards,

Alok Ranjan

Academic Editor

PLOS ONE
---

## [Editor Report · Acceptance letter]

5 Jan 2023

PONE-D-21-35310R2 

Factors affecting motivation of close-to-community sexual and reproductive health workers in low-income urban settlements in Bangladesh: a qualitative study 

Dear Dr. Mahmud:

I'm pleased to inform you that your manuscript has been deemed suitable for publication in PLOS ONE. Congratulations! Your manuscript is now with our production department. 

Kind regards, 

on behalf of

Dr. Alok Ranjan 

Academic Editor

PLOS ONE